# Isolation and Characterization of Extracellular Vesicles from Gastric Juice

**DOI:** 10.3390/cancers14143314

**Published:** 2022-07-07

**Authors:** Gleb O. Skryabin, Svetlana V. Vinokurova, Sergey A. Galetsky, Danila S. Elkin, Alexey M. Senkovenko, Darya A. Denisova, Andrey V. Komelkov, Ivan S. Stilidi, Ivan N. Peregorodiev, Olga A. Malikhova, Oiatiddin T. Imaraliev, Adel D. Enikeev, Elena M. Tchevkina

**Affiliations:** 1Institute of Carcinogenesis, N. N. Blokhin National Medical Research Center of Oncology, Kashirskoye Sh. 24, 115478 Moscow, Russia; gleb-skryabin@mail.ru (G.O.S.); vinokourova@mail.ru (S.V.V.); sg8126@mail.ru (S.A.G.); yodanila@yandex.ru (D.S.E.); darhance@yandex.ru (D.A.D.); adelbufyeni@mail.ru (A.D.E.); 2Department of Bioengineering, Faculty of Biology, M.V. Lomonosov Moscow State University, Leninskie Gory, 1/12, 111234 Moscow, Russia; senkovenkoam@my.msu.ru; 3Research Institute of Clinical Oncology, N. N. Blokhin National Medical Research Center of Oncology, Kashirskoye Sh. 24, 115478 Moscow, Russia; director@ronc.ru (I.S.S.); i.peregorodiev@ronc.ru (I.N.P.); malikhova@inbox.ru (O.A.M.); o.imaraliev@ronc.ru (O.T.I.)

**Keywords:** extracellular vesicles, exosomes, gastric cancer, gastric juice, miRNA, stomatin, cancer markers

## Abstract

**Simple Summary:**

Gastric cancer (GC) is one of the most common cancers and the fifth leading cause of cancer-related deaths worldwide. The steadily growing interest in secreted extracellular vesicles (EVs) is related to their ability to carry a variety of biologically active molecules, which can be used as markers for liquid noninvasive diagnosis of malignant neoplasms. For these applications, blood is the most widely used source of EVs. However, this body fluid contains an extremely heterogeneous mixture of EVs originating from different types of normal cells and tissues. The aim of this study was to assess the possibility of using gastric juice (GJ) as an alternative source of EVs since it is expected to be enriched in vesicles of tumor origin. We validated the presence of EVs in GJ using transmission electron microscopy (TEM), nanoparticle tracking analysis (NTA) and western-blot analysis of exosomal markers, showed for the first time the feasibility of their isolation by ultracentrifugation and demonstrated the prospect of using GJ-derived EVs as a source of GC miRNA markers.

**Abstract:**

EVs are involved in local and distant intercellular communication and play a vital role in cancer development. Since EVs have been found in almost all body fluids, there are currently active attempts for their application in liquid diagnostics. Blood is the most commonly used source of EVs for the screening of cancer markers, although the percentage of tumor-derived EVs in the blood is extremely low. In contrast, GJ, as a local biofluid, is expected to be enriched with GC-associated EVs. However, EVs from GJ have never been applied for the screening and are underinvestigated overall. Here we show that EVs can be isolated from GJ by ultracentrifugation. TEM analysis showed high heterogeneity of GJ-derived EVs, including those with exosome-like size and morphology. In addition to morphological diversity, EVs from individual GJ samples differed in the composition of exosomal markers. We also show the presence of stomatin within GJ-derived EVs for the first time. The first conducted comparison of miRNA content in EVs from GC patients and healthy donors performed using a pilot sampling revealed the significant differences in several miRNAs (-135b-3p, -199a-3p, -451a). These results demonstrate the feasibility of the application of GJ-derived EVs for screening for miRNA GC markers.

## 1. Introduction

Extracellular vesicles are phospholipid bilayer membrane-enclosed particles of different sizes and intracellular origin secreted by various cells.

The process of EV formation is closely related to the selection and loading of biologically active molecules, which are further transported into recipient cells or interact with plasma membrane molecules. This ultimately leads to epigenetic changes and alterations in intracellular signaling [1]. Although EV secretion is shown for almost all cell types, a large body of evidence suggests that cancer cells release higher amounts of EVs compared to non-malignant cells [2,3]. Moreover, the EV secretion seems to increase with cancer progression and the disease stage [4,5,6]. Furthermore, given the high stability of EV cargo molecules as well as the detection of vesicles in all body fluids, EVs are considered a promising source of cancer markers [7,8,9].

Gastric cancer is one of the most common cancers worldwide and is responsible for over one million new cases in 2020 and an estimated 769,000 deaths, ranking sixth for incidence and fifth for mortality globally [10]. GC represents a highly heterogeneous group of diseases, 95% of which are adenocarcinomas [11]. Despite the progress in surgery and therapy techniques, the 5-year survival rate of GC patients is still low, mostly due to the low rate of early diagnosis [12]. A high proportion of advanced cancer stages as well as high morphogenetic variability of GC determine the requirement of additional markers for early diagnosis and differential diagnosis. Secreted extracellular vesicles are of growing interest in terms of screening for new molecular markers for noninvasive liquid cancer diagnosis.

Blood (plasma or serum) is the most commonly used body fluid for EV analysis. EVs from peritoneal lavages and ascitic fluids are also used to search for gastric cancer markers [13,14,15]. It should be noted that blood as a source of EVs for the study of cancer markers has several limitations, the main ones being the extremely high overall heterogeneity of vesicles and the low proportion of EVs of tumor origin [16,17]. In addition, blood is enriched with particles of non-vesicular origin with sizes and other physical characteristics (such as lipoprotein complexes) similar to EV, which contaminate EV preparations [18].

Peritoneal fluid probably contains a higher proportion of vesicles of tumor origin, although this body fluid is more suitable as a source of markers for prognosis assessment or tumor staging than for diagnosis. GJ appears to be a promising source of EVs for the search for diagnostic markers of GC since it can be obtained from both healthy people and cancer patients, and in the case of GC, it should be enriched with vesicles of tumor origin. To our surprise, EVs from GJ have been largely unexplored. The presence of EVs in GJ has been shown in two early studies [19,20] in which, however, EVs were not sufficiently characterized according to ISEV (International Society for Extracellular Vesicles) guidelines. In 2019, Kagota et al. clarified the existence of EVs in GJ [21]. The authors of the latter study developed a significantly modified protocol since they failed to isolate EVs using the standard ultracentrifugation method. This protocol contained two additional steps, including a rather complicated preprocessing procedure and the binding of ultracentrifuged vesicles to microbeads. Accordingly, the morphology of Evs has not been properly studied.

The objectives of this study were to test whether EVs can be isolated from GJ using the standard ultracentrifugation technique; to determine the concentration of GJ-derived EVs in the obtained preparations; to characterize the obtained EVs by size and morphology and to evaluate the presence and composition of various exosomal markers according to the criteria recommended by ISEV; to compare the content of certain miRNAs in EVs from GC patients and healthy subjects through a pilot sample.

We demonstrate the feasibility of the ultracentrifugation method for the isolation of EVs, including exosome-like vesicles, from GJ. The high yield of EVs allows for performing further analysis of their molecular cargo. Notably, the content of proteins commonly used as exosomal markers, such as CD9, Alix, TSG-101, Flot-2, varied enormously in EVs obtained from different individuals. Moreover, the size spectra of CD9(+) and CD9(−) EV samples differed significantly, indicating the existence of distinct subtypes of vesicles present in GJ. Interestingly, the majority of the EV samples contained stomatin, a member of the SPFH family, previously suggested as a new EV marker [22]. First-time analysis of miRNA content in GJ-derived EVs from GC patients and non-cancer individuals revealed significant upregulation of miR-135b-3p and miR-199a-3p and downregulation of miR-451a.

## 2. Materials and Methods

### 2.1. Clinical Specimens and Patient Consent

Gastric juice samples were obtained from gastric cancer patients (intermediate to high-grade adenocarcinoma, GC patients, *N* = 7; clinical and morphological characteristics are shown in Appendix A, Appendix A) and patients with no history of cancer (control group, *N* = 6) using a OLYMPUS GIF H-185 diagnostic videogastroscope at the beginning of the endoscopy. All patients were food-starved for 12 h and water-starved for 6 h before manipulation. Samples were received from the Endoscopy Department of the N.N. Blokhin National Medical Research Center of Oncology. Written informed consent was sought and obtained from all participants in accordance with the N.N. Blokhin National Medical Research Center of Oncology Ethics Committee guidelines.

### 2.2. Sample Processing

The initial volume of GJ samples ranged from 2 to 5 mL. The obtained samples were diluted with 5 mL of ice-cold PBS (#70011-044, Gibco, Grand Island, NY, USA) just after collection and processed within 2 h at 4 °C. Further steps of sample processing were performed according to the method described by Théry et al. for purifying exosomes from viscous fluids [23] with slight modifications. After brief vortexing, samples were centrifuged at 800× *g* for 20 min and at 2000× *g* for 30 min using an A-4-81 rotor (Eppendorf Centrifuge 5810R, Eppendorf AG, Hamburg, Germany) to remove cells and loose cellular mucosal debris. It is noteworthy that sometimes white-yellowish flakes remained in the non-transparent supernatant obtained. They were sedimented alongside large particles at 12,000× *g* using an F-34-6-38 rotor (Eppendorf Centrifuge 5810R) for 1 h after four-times dilution with ice-cold PBS. After these steps, transparent supernatants were frozen at −80 °C until further steps of EV isolation.

### 2.3. Isolation of EVs

We followed the protocol for EV isolation from viscous fluids by differential ultracentrifugation described by Théry et al. [23] and Caby et al. [24] with slight modifications. Thawed supernatants were diluted with ice-cold PBS to a final volume of 35 mL and transferred to ultracentrifuge tubes (#326823, Beckman Coulter, Brea, CA, USA) to perform a first ultracentrifugation round at 110,000× *g* (4 °C) for 3 h using an SW-28 swinging bucket rotor (*k* factor 245.5; Beckman Coulter). The obtained pellets (containing mostly small EVs) were resuspended in 5 mL of ice-cold PBS (Gibco), transferred to small ultracentrifuge tubes (#326819, Beckman Coulter) and centrifuged again at 110,000× *g* (4 °C) for 90 min using an SW-50.1 swinging bucket rotor (*k* factor 154.5; Beckman Coulter). The final cleared pellets were resuspended in 120 μL of ice-cold PBS and aliquoted in Protein LoBind tubes (#0030108434, Eppendorf AG, Hamburg, Germany) for NTA, TEM, protein analysis and RNA extraction. Aliquots were frozen in liquid nitrogen and stored at −80 °C for further analysis.

### 2.4. Particle Size Distribution and Quantification

Size distribution and concentration of EVs were determined by NTA using a NanoSight LM10 HS instrument equipped with a NanoSight LM14 unit with on-board temperature control (Malvern Panalytical Ltd., Malvern, UK), LM 14C (405 nm, 65mW) laser unit and high sensitivity camera with a Scientific CMOS sensor (C11440-50B, Hamamatsu Photonics, Hamamatsu City, Japan). Six 60 sec videos were recorded for two independent replicates, generating 12 individual measurements for each sample. Further processing was performed as we described previously [25].

### 2.5. Transmission Electron Microscopy

EV’s morphology analysis was performed using a JEM-1011 transmission electron microscope (JEOL, Ltd., Akishima, Japan) operating at 80 kV according to the protocol described in Skryabin et al. (at least 10 fields of view per sample) [25].

### 2.6. Immunoblotting and Antibodies

The concentration of total protein in EV samples and cells lysed in RIPA buffer was determined using the NanoOrange^™^ protein quantitation kit (#N6666, ThermoFisher Scientific, Eugene, OR, USA) according to the manufacturer’s recommendations using a SpectraMax M5e microplate reader (Molecular Devices, LLC., San Jose, CA, USA). Immunoblotting was performed according to the previously described procedure [25] with the differences that 5 µg of total protein was applied to SDS-PAGE and proteins were visualized with SuperSignal^™^ West Femto Maximum Sensitivity Substrate (#34095, ThermoFisher Scientific, Rockford, IL, USA). The following primary and secondary antibodies and dilutions were used: anti-Alix (#sc-271975, 1:500; Santa Cruz Biotechnology, Dallas, TX, USA), anti-Flotillin-2 (#3436S, 1:1000; Cell Signaling Technology, Topsfield, MA, USA), anti-CD9 (#13174, 1:2000; Cell Signaling Technology), anti-TSG-101 (ab125011, 1:5000; Abcam, Cambridge, UK), anti-PCNA (#sc-7907, 1:500; Santa Cruz Biotechnology), anti-Stomatin (#sc-134554, 1:500; Santa Cruz Biotechnology), anti-mouse goat polyclonal antibodies (#2367, 1:5000; Cell Signaling Technology); and anti-rabbit goat polyclonal antibodies (#29902, 1:80,000; Cell Signaling Technology).

### 2.7. RNA Isolation, Reverse Transcription and Quantitative Real-Time PCR

RNA from EVs was isolated using the Total Exosome RNA and Protein Isolation Kit (#4478545; ThermoFisher Scientific, Vilnius, Lithuania) according to the manufacturer’s protocol. RNA was eluted from the last column with 60 µL of nuclease-free water and stored at −80 °C until further analysis. Concentration, size distribution and percentage of small RNA were analyzed by Agilent 2100 Bioanalyzer using Small RNA Kits (Agilent Technologies, Santa Clara, CA, USA).

Stem-loop RT-PCR for miRNA quantification was performed according to the method described by Chen et al. [26]. RNA concentration was measured using a NanoDrop™ ND-1000 Spectrophotometer (ThermoFisher Scientific, Wilmington, DE, USA), and 6 ng of exosomal RNA was used for stem-loop RT-PCR according to the previously described procedure [25].

Primers for specific miRNAs were designed using miRBase v22.1 and synthesized by “DNA-synthesis Ltd.” (Moscow, Russia) (sequences are shown in Appendix A, Appendix A). The amplification efficiencies were tested according to the previously described protocol [25]. Melting temperatures were 59 °C for miR-199a-3p, miR-204-3p, miR-16-5p and let-7b-5p; 54 °C for miR-451a; and 60 °C for miR-23a-3p and miR-135b. In the case of hsa-miR-135b (-3p and -5p), TaqMan^™^ miRNA Assay (Assay ID: 002159, 002261; ThermoFisher Scientific) was used according to the manufacturer’s protocol.

miRNA expression data were normalized to miR-23a-3p. Fold change (FC) was determined using the ΔΔCt method, where ΔCt = Ct(miRNA) − Ct(miR-23a) and ΔΔC(t) = ΔCt(sample) − average ΔCt(control), and FC = 2^−ΔΔCt^.

### 2.8. Statistical Analysis

Based on the NTA-measured particle size and concentration, values of mean, mode, percentile data (10th and 90th), standard deviation, and confidence interval were calculated using Wolfram Mathematica ver. 11 ((Wolfram Research, Champaign, IL, USA) software. Student’s *t*-test and analysis of variance (ANOVA) were used for the comparison of groups. *p*-values lower than 0.05 were considered statistically significant. For statistical analysis, we used the statistical software package GraphPad Prism ver. 8.0.0 package for MS Windows, engineering-mathematical package Wolfram Mathematica ver. 11. The package MS Excel 2016 for MS Windows was used for plotting graphs.

## 3. Results

### 3.1. Characterization of EVs Isolated from Gastric Juice

GJ samples were collected from patients with gastric cancer (*N* = 7, clinical and morphological characteristics are shown in Appendix A, Appendix A) and individuals without a history of cancer (non-cancer patients, *N* = 6) during a routine esophagogastroduodenoscopy procedure. EVs were isolated using an ultracentrifugation-based method with slight modifications (see Materials and Methods). The size and concentration of EVs were assessed by NTA (Figure 1A–C). The levels of various exosomal markers were tested by immunoblotting (Figure 2A). EV size and morphology have been visualized by TEM (Figure 2B).

All preparations obtained from GJ contained a high number of “cup-shape” particles corresponding in size and morphology to EVs under the TEM analysis.

The mean size of particles assessed by NTA varied from 92 to 178 nm, with modes from 52 to 141 nm in different individual EV preparations. The mean size of EVs and median over the entire sampling were 149 (SD 29) and 133 nm (SD 37), correspondingly (Figure 1C). According to NTA data, the concentration of EVs varied from 10^11^ to 10^13^ particles per mL (the average EV concentration over the entire sampling was 5.12 × 10^12^ particles/mL). The mean size of the EVs in preparations obtained from GC patients and individuals without a history of cancer (non-cancer patients) had no significant differences (Student’s *t*-test, *p* > 0.05). Furthermore, no relationship between vesicle characteristics and tumor grade was observed (*p* > 0.05).

To confirm the presence of exosomes in EV preparations, we further studied the exosomal markers in EV preparations by immunoblotting. In accordance with ISEV guidelines [27], several proteins from different functional classes and with different intracellular compartmentalization were analyzed. The selected markers included tetraspanin CD9, the tumor susceptibility gene protein 101 (TSG-101) and Alix, known members of the ESCRT (endosomal sorting complex required for transport)-dependent pathway of exosome biogenesis; flotillin-2, a structural and functional component of membrane microdomains, as well as stomatin, which we proposed previously as a new exosomal marker [22]. PCNA was used to confirm the absence of cellular proteins of non-vesicular origin in EV preparations. Cell lysate of the gastrointestinal stromal tumor cell line, GIST-T1, and GC tissue lysate were used for the comparison of protein levels in EVs and cells. All proteins were analyzed in a single experiment, making it possible to compare the ratio of studied proteins in different EV preparations. The results of the analysis showed high variability in the composition of exosomal markers among individual EV preparations. In particular, we observed CD9-positive EVs devoid of all or some of the other markers; CD9-negative samples enriched with other markers, and EVs containing all the markers studied (Figure 2A). To confirm the presence of EVs in samples with different combinations of exosomal markers, Figure 2B shows EV images from TEM analysis of the respective samples.

To find out whether there are differences between vesicles containing different sets of exosomal markers, we compared their size distributions. We found that the mean size of vesicles, as well as mode value (the particle size that appears most often in a set of data values) for CD9-positive and CD9-negative EVs, had significant differences (*p* < 0.01) (Figure 1D). Thus, the average size and the mode value for CD9(+) EVs were 165 and 106 nm, and for CD9(−) EVs—116 and 58 nm, respectively. It is likely that EVs with different compositions of exosomal markers, in particular CD9(+) and CD9(−) vesicles, represent different subtypes of EVs, which may or may not be present in various combinations in individual GJ samples.

Remarkably, in addition to the full-length CD9 protein present in both cell lysates and most EV samples, a lower molecular weight protein was observed in EVs exclusively. We speculate that it corresponds to a fragmented CD9, resulting from the proteolytic activity of gastric proteases such as pepsin or gastricin, which could lead to the cleavage of tetraspanin domains located on the outer side of the EV membrane.

Notably, we showed for the first time the presence of stomatin protein in almost all GJ-derived EV samples. This finding confirms our previously published data linking this protein to the biogenesis of EVs [22]. Stomatin has never been studied in EVs except for EVs originating from blood cells. Another interesting observation is the presence of vesicles with non-canonical morphology, including elongated and multilayered ones (Figure 3). Although, it cannot be ruled out that this shape is caused by distortions resulting from TEM analysis. Vesicles of similar morphology have been shown previously in several studies [28,29,30].

### 3.2. miR-135b-3p, miR-199a-3p and miR-451a Are Differently Presented in EVs from GJ of Gastric Cancer Patients and Non-Cancer Individuals

Data from the Agilent 2100 Bioanalyzer revealed the wide spectrum of small RNAs presenting in GJ-derived EVs, including a peak of about 23 nucleotides corresponding to microRNAs. The percentage of microRNA in total small RNA varied drastically, reaching the maximum of 43%, as shown in Figure 4A.

miRNA levels in EVs from GJ of GC patients and non-cancer individuals were further analyzed by stem-loop RT-qPCR. Several miRNAs were selected for the initial study based on literature data, including hsa-miR-135b-3p, hsa-miR-135b-5p, hsa-miR-199a-3p, hsa-miR-204-3p, hsa-miR-451a, hsa-miR-16-5p and hsa-let-7b-5p. Sequences of primers used for individual miRNA analysis are shown in Appendix A, Appendix A. According to RT-qPCR data, one of the most equally expressed miRNAs in all EV samples was miR-23a-3p. As the levels of miR-16-5p as well as let-7b-5p, often used for miRNA normalization, varied significantly between EV samples, we used miR-23a-3p as a reference to assess miR-135b-3p, miR-135b-5p, miR-199a-3p, miR-204-3p and miR-451a (Ct values for each sample are shown in Appendix A, Appendix A). The difference between studied miRNA levels in the compared groups (fold change) was calculated using the ΔΔCt method. We found miR-135b-3p and miR-199a-3p to be significantly upregulated (4.29- and 3.97-fold increase, respectively) and miR-451a to be downregulated (3.78-fold decrease) in EVs from GC patients compared to the control group (*p* < 0.05) (Figure 4B). The level of miR-204-3p showed no significant difference between compared groups. Surprisingly, there were also no significant differences in miR-135b-5p expression, although upregulation of this miRNA in gastric cancer has been shown repeatedly. However, it should be noted the wide scattering of Ct values for this miRNA in EV specimens.

These results indicate a different composition of miRNAs in EVs obtained from GJ of GC patients and non-cancer individuals. However, the data on the differential expression of certain miRNAs obtained in this initial study should be interpreted with caution due to the strong variability in EV composition identified here. Further studies are needed to understand the reasons for this heterogeneity as well as to determine differences in the molecular composition of distinct subpopulations of vesicles characterized by different sets of exosomal markers.

## 4. Discussion

Exosomes and microvesicles belong to the secreted EVs, a heterogeneous group of cell-derived membrane structures. EVs are involved in intercellular communication through the exchange of cargo biomolecules, consisting of proteins, lipids, metabolites and various types of nucleic acids. They are present in almost all body fluids and participate in multiple physiological and pathological processes mediating epigenetic regulation of gene expression and alterations in intracellular signaling [31,32].

The results of numerous studies indicate that EVs contribute to the malignant phenotype and the survival of primary tumor cells, regulating the processes of proliferation and apoptosis [33,34]. EVs also participate in tumor spread by enhancing the migratory and invasive activity of cancer cells and stimulation of angiogenesis [35,36,37,38]. In addition, EVs secreted by tumor cells are involved in the processes of the tumor–stroma interaction, reorganization of tissue microenvironment, pre-metastatic niche formation, and reprogramming of immune cells to evade anti-tumor immunity [7]. Repeatedly described similarities in molecular signatures, including proteome and transcriptome associations, between parental cells and secreted EVs have highlighted the potential of tumor-derived EV molecules as promising liquid biopsy markers for cancer diagnosis and monitoring.

The importance of EVs in the pathogenesis of gastric cancer has been confirmed by numerous studies on both experimental models [39,40] and clinical specimens [41]. For instance, several studies have shown that GC-derived exosomes promote tumor cell proliferation and invasion through activation of PI3K/Akt and MAPK/ERK-dependent signaling pathways [42,43]. Exosomes have also been shown to be involved in Treg cell formation through TGF-β1 activity and contribute to lymphogenic metastasis of gastric carcinoma [44,45]. Several data indicate that exosomes participate in mesothelial-to-mesenchymal transition and promote peritoneal metastasis, a primary metastatic route in advanced GC [46,47,48]. Zhang et al. demonstrated that exosomes promote GC liver metastasis through the delivery of EGFR and rearrangement of the liver microenvironment [49].

The association of EV molecular composition with tumor malignancy and cancer progression has been repeatedly shown for many cancer types, including GC [50,51,52,53,54]. Furthermore, based on deep sequencing data, differences in miRNA expression profiles in exosomes derived from GC stem-like cells and differentiated GC cells were shown [55]. Numerous studies have focused on the identification of EV markers for the diagnosis and prognosis of gastric cancer [56,57]. Based on the differences in proteomic profiles of exosomes from the serum of GC patients and healthy controls, down-regulation of TRIM3 protein was suggested as a biomarker for GC diagnosis [58]. Similarly, a decrease in exosomal gastrokine-1 level has been shown to be associated with gastric cancer [59]. Among the miRNAs proposed as GC markers were miR-101, which has been significantly reduced in both exosomes and plasma of GC patients [60], and miR-23b, the level of which in plasma-derived exosomes was associated with recurrence and progression of GC [61]. Wang et al. identified a panel of serum exosomal miRNAs, including miR-19b-3p, miR-17-5p, miR-30a-5p, and miR-106a-5p for GC diagnosis [62]. Several studies demonstrate the potential significance of certain long noncoding RNAs such as LINC00152 and HOTTIP in relation to GC diagnosis and prognosis [63,64].

The above examples demonstrate the high potential of exosomes as a source of GC biomarkers. However, it is worth noting the low convergence of the data with respect to the specific molecules identified in the various studies. In addition to differences in methodological approaches to the isolation of EVs and analysis of their molecular composition, the inconsistency of the data can be explained by the choice of blood (plasma or serum) as a source of EVs. Blood is an extremely heterogeneous body fluid in terms of EV composition, in which the vast majority of EVs are produced by blood cells, immune cells and epithelial cells of different histogenesis, while only a very small percentage of vesicles are of tumor origin [17]. In addition, the composition of exosomes in the blood differs rather unpredictably according to a variety of factors, including gender, age, lifestyle and many other parameters [16].

In some studies, malignant ascites and peritoneal fluid have been used as a source of EVs. Such body fluids might contain a higher proportion of tumor-derived EVs and thus more fully reflect tumor-associated changes in EV composition. At the same time, this approach seems more effective for detecting prognostic markers and assessing recurrence than for diagnostic tasks. For example, several exosomal miRNAs from peritoneal fluid have been shown to be associated with peritoneal metastasis, including four miRNAs (miR-21-5p, miR-92a-3p, miR-223-3p and miR-342-3p) that were elevated and miR-29 family members that were decreased in patients with peritoneal metastases [13]. Based on the analysis of exosomes from GC malignant ascites, peritoneal lavage fluids, and conditioned media of GC cell lines, miR-21 and miR-1225-5p were identified as potential prognostic biomarkers of peritoneal metastasis [14]. In addition, reduced expression of exosomal miR-29 family in peritoneal fluid has also been shown as a predictor of peritoneal recurrence of GC [15].

GJ appears to be a very suitable source of EVs for the task of searching for diagnostic markers of GC, as the expected proportion of EVs originating from tumor cells and microenvironmental cells should be significantly higher in this body fluid compared to the circulation. In addition, unlike blood, GJ should not contain ribonucleoproteins and lipoprotein complexes, which almost inevitably contaminate EV preparations. Surprisingly, GJ-derived EVs have been hardly investigated so far, with the exception of the few above-mentioned studies [19,20,21].

We confirmed the presence of EVs in GJ and showed that they can be isolated by ultracentrifugation-based techniques. The obtained EVs were characterized according to ISEV recommendations [27], including the size and morphology of vesicles (determined by NTA and TEM), as well as the expression of exosomal markers belonging to different functional protein groups with different intracellular localization. TEM analysis revealed remarkable morphological heterogeneity of GJ-derived EVs. Particularly, unlike the EVs we observed in other sources, such as ascitic fluid, blood plasma, uterine lavage, or cell-conditioned media [22,65,66], preparations from GJ contain vesicles, very similar in shape to those previously described in several papers, such as so-called “double”, “tubular” and “multilayered” vesicles [28,29,30]. It cannot be ruled out that such shapes are artifacts resulting from overlapping vesicles or other distortions caused by technical problems. Further studies using cryo-EM and other imaging techniques are needed to clarify this issue.

Another interesting observation is the high variability in the content of exosomal markers in GJ-derived EVs. Thus, tetraspanin CD9 was detected in 9 out of 13 samples; cytosolic proteins, including TSG-101 and Alix (components of the ESCRT complex)—in 12 and 7 out of 13 EV preparations, respectively; the membrane protein flotillin-2 (a component of lipid microdomains)—in 10 out of 13 EV preparations. It is noteworthy that the indicated proteins were present in almost all combinations. That means, apparently, that EVs isolated from GJ by ultracentrifugation consist of different subpopulations of vesicles characterized by a distinct set of exosomal markers.

Variability in the composition of exosomal markers among different EV populations, presented in both body fluids and cell culture media, has been shown repeatedly. It is still not entirely clear what exactly accounts for these differences, and the data from various studies in this regard are rather contradictory. For instance, comparative proteomic analysis has identified four subcategories even in the category of small vesicles, namely: sEVs coenriched in CD63, CD9, and CD81 tetraspanins and endosome markers; sEVs devoid of CD63 and CD81 but enriched in CD9; sEVs devoid of CD63/CD9/CD81; and sEVs enriched in serum- or extracellular matrix-derived factors [67]. In contrast to these findings, another study states that sEVs bearing CD9 and CD81 with little CD63 correspond to ectosomes, whereas others bearing CD63 with little CD9 were qualified as exosomes [68]. Using Rab27a inhibition to modulate exosome secretion, Bobrie et al. showed the existence of at least two distinct populations of sEVs, the secretion of which was differently dependent on Rab27a, that is, the Rab27a-dependent subpopulation containing CD63, TSG-101, Alix and Hsc70, and the Rab27a-independent one enriched in CD9 and Mfge8 [69]. In contrast, based on a comparison of the protein content of EVs from 60 cell lines, it was shown that only CD81, Alix, and HSC70 were present across all samples, while other proteins, including CD63, CD9, TSG-101, syntenin-1, and flotillin-1, were present in at least two-thirds of the samples [70].

Such heterogeneity in the data can be attributed both to the natural heterogeneity of the vesicles and to the diversity of isolation methods, which may result in the enrichment of preparations with different subpopulations of vesicles [71] or particles of non-vesicular origin [72]. The association of exosomal marker composition with different vesicle subpopulations is also confirmed here by the revealed correlation between vesicle size and the presence of CD9. We suggest that CD9(−) vesicles may represent a distinct population of smaller EVs.

Another noteworthy feature is the presence of two forms of CD9 revealed by immunoblotting. The first one of 24 kDa corresponds to a full-length protein and is present in both cell lysates and in EVs, while the other is a lower molecular-weight (approximately 20 kDa) protein of unknown origin. We have not previously observed CD9 of this size in EVs from other origins, including blood plasma, ascites, aspirates and flushes from uterine cavity, culture medium, etc. [22,25]. We hypothesize that the truncated CD9 results from the proteolytic activity of gastric juice proteases, such as pepsin or gastricsin, which may lead to cleavage of the tetraspanin’s extracellular domains located on the outer surface of the vesicle membrane.

Among other proteins presented in GJ-derived EVs we found stomatin. Stomatin, similarly to flotillins, belongs to the SPFH (stomatin/prohibitin/flotillin/HflK/C) protein family and colocalizes with flotillins in lipid microdomains (also called lipid rafts) [22,73,74], highly dynamic liquid-ordered subdomains enriched in sterols, sphingolipids and glycosphingolipids and involved in the compartmentalization of signaling and transport processes on the plasma membrane [75]. Although several data indicate that lipid rafts play an important role in the formation of intraluminal vesicles (intracellular precursors of exosomes) [76], stomatin has not previously been studied in relation to EVs. In our previous study, we first showed the presence of stomatin in exosomes produced by epithelial cancer cells (lung, breast and ovarian cancer cells) as well as in EVs from various body fluids, including blood plasma, ascitic fluid and uterine flushes. Based on the high content of stomatin in EVs of various origins and its enrichment in exosomes, we proposed this protein as a promising exosomal marker [22]. Here, we observed stomatin in almost all EV samples, confirming its ubiquitous presence in EVs of various origins and indicating its association with EV biogenesis.

Next, we confirmed the presence of various miRNAs in GJ-derived EVs by RT-qPCR. Several microRNAs were selected for the pilot analysis based on literature data, including miR-135b-3p, miR-199a-3p, miR-204-3p, miR-451a. Levels of miR-135b-3p were found to be significantly higher in EVs from GC patients compared to non-cancer individuals. This result confirms numerous data on the tumor-promoting role of miR-135b, both miR-135b-3p and miR-135b-5p forms in GC carcinogenesis. miR-135b is involved in epithelial-mesenchymal transition, proliferation, apoptosis, migration, angiogenesis, and anticancer immunity through the regulation of a number of intracellular signaling pathways, including MAPK and PI3K/AKT cascades, FOXO, Wnt, and TGFβ-dependent pathways and several others [77,78,79,80,81]. The involvement of miR-135b in GC progression is also mediated through its intercellular transmission within the exosomal cargo [82,83]. It has also been shown that the expression level of miR-135b in plasma of GC patients is generally higher than in healthy individuals. Moreover, an increase in miR-135b has been proposed as a prognostic marker [81] as well as a diagnostic marker of GC [84]. This oncomiR was also among seven miRNAs identified as robust biomarkers for GC by a bioinformatic integrated analysis of differentially expressed miRNAs from five microarray datasets in the Gene Expression Omnibus database. In this study, seven miRNAs were filtered from fourteen primary miRNAs using the validation set of The Cancer Genome Atlas Program database [85].

Among the miRNAs identified in the same study was miR-204, the downregulation of which in GC has been suggested as a marker for early diagnosis of GC. At the same time, the results of our study showed no significant differences in miR-204-3p levels between GJ-derived EVs from GC and non-cancer samples. miR-199a-5p, according to the same study, has been classified as one of the most frequently altered in cancer but was categorized as a microRNA “with unclear expression changes in GC tissues compared to normal tissues or adjacent normal tissues”. Indeed, the deregulation of miR-199a-5p in tumors and its involvement in carcinogenesis has been shown in many in vitro and in vivo studies [86]. In gastric cancer tissues, miR-199a-3p expression was shown to be upregulated in 69.2% of patients [87]. Consistent with these results, we found an increased content of miR-199a-3p in EVs from patients with GC compared to non-cancer patients. In the same samples, we found a significant decrease in miR-451a, indicating that this miRNA may act as a tumor suppressor. The negative role of this miRNA in cancer progression is well established. For example, Streleckiene et al. found that miR-451a is markedly deregulated and displays tumor-suppressive activity in GC through regulation of the PI3K/AKT/mTOR signaling pathway [88]. Su et al. showed decreased miR-451 expression in the GC tissues and cell lines and reported that downregulation of miR-451 tended to be positively correlated with lymphatic metastasis, advanced clinical stage, and shorter overall survival in patients with GC [89]. Shen et al. demonstrated a correlation of low miR-451 expression with tumor stage, lymphatic metastasis, and overall survival in patients with GC and suggested the downregulation of miR-451 as a diagnostic and prognostic biomarker in GC [90]. Similar results evidencing the prognostic significance of miR-451a have also been reported based on the investigation of tumor tissues and the clinicopathological features of 180 patients with GC [91].

## 5. Conclusions

In conclusion, we have shown for the first time that exosome-like EVs can be isolated from GJ by ultracentrifugation in amounts sufficient for further analysis of their molecular cargo, including miRNA composition. Analysis of exosomal protein markers revealed differences in size between CD9(+) and CD9(−) EV populations, indicating the existence of distinct subtypes of EVs in GJ. The results of the analysis of a pilot sampling of EVs showed a significant increase in miR-135b-3p and miR-199a-3p, as well as a decrease in miR-451a levels in EVs from GC patients compared to non-cancer individuals. The observed changes indicate for the first time the difference in the content of miRNAs in EVs present in GJ of GC patients and healthy individuals. Thus, EVs derived from GJ are a promising source of miRNA markers of gastric cancer.

## Figures and Tables

**Figure 1 cancers-14-03314-f001:**
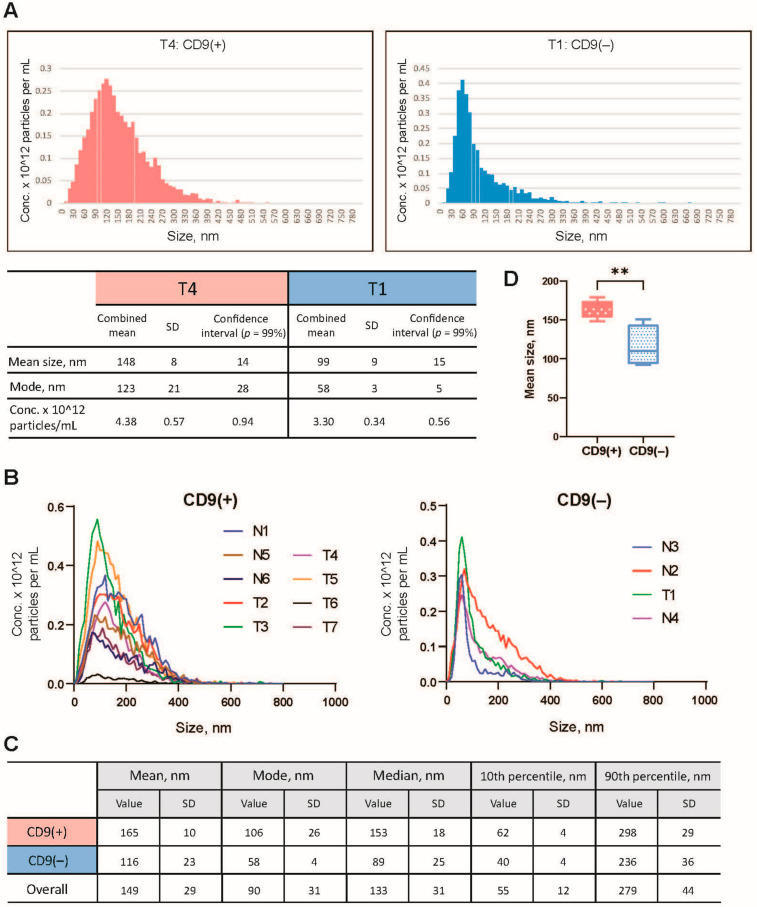
The characterization of GJ-derived EVs by nanoparticle tracking analysis. T1–T7—samples obtained from GC patients. Clinical and morphological characteristics are shown in Appendix A, Appendix A. N1-N6—samples obtained from non-cancer individuals. (**A**) Examples of size spectra of a CD9-positive (CD9(+)) and a CD9-negative (CD9(−)) EV samples isolated from GJ of GC patients (samples T1, T4). (**B**) NTA data for EV size distribution among all CD9(+) and CD9(−) EV samples studied. (**C**) Main NTA characteristics of CD9(+) EVs, CD9(−) EVs and across the entire sample. (**D**) Comparison of the mean size of CD9(+) and CD9(−) EVs (** *p* < 0.01).

**Figure 2 cancers-14-03314-f002:**
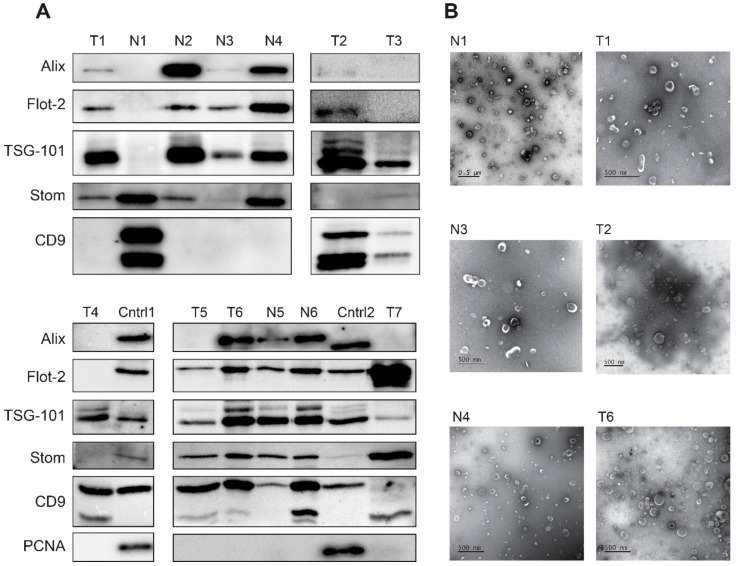
Analysis of EV morphology and exosomal marker composition. (**A**) Western-blot analysis of exosomal markers Alix, flotillin-2 (Flot-2), TSG-101, and CD9 as well as stomatin protein (Stom) in EVs from GJ of GC patients (T1–T7, clinical and morphological characteristics are shown in Appendix A, Appendix A) and non-cancer individuals (N1–N6). Full Western blot images can be found in Appendix A. The PCNA protein was used to confirm the absence of cellular proteins of non-vesicular origin in EV preparations. Protein lysates of GIST-T1 cells (Cntrl 1) and GC tissue (Cntrl2) were used as molecular weight controls and to compare levels of proteins in cells and EVs. Two bands of CD9 protein correspond to full-size form (24 kDa) and lower molecular weight form (of about 20 kDa). (**B**) TEM analysis of EV morphology. Examples of CD9(+) EVs (samples N1, T2, T6) and CD9(−) EVs (samples T1, N3, N4) isolated from GJ of GC patients (T) and non-cancer individuals (N); scale bar 500 nm.

**Figure 3 cancers-14-03314-f003:**
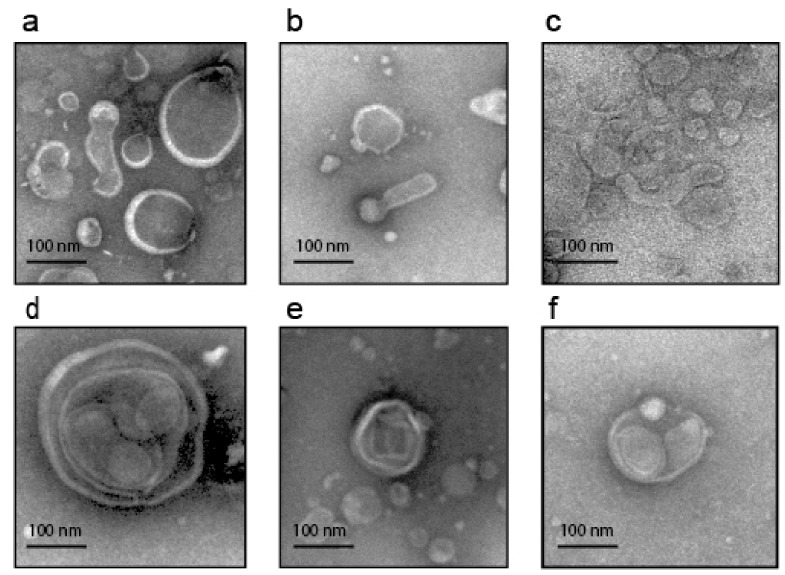
Examples of vesicles with atypical morphology visualized by TEM: (**a**–**c**)—elongated (tubular) EVs; (**d**–**f**)—multilayered EVs. The scale bars correspond to 100 nm.

**Figure 4 cancers-14-03314-f004:**
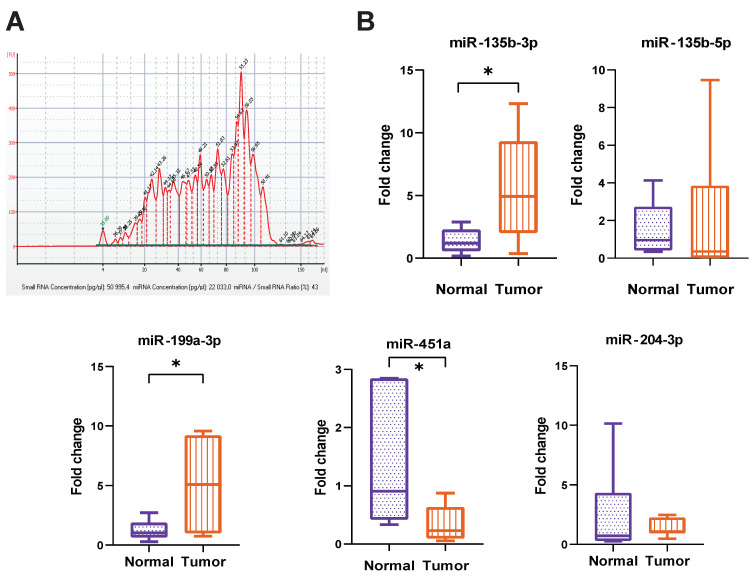
Analysis of miRNA levels in EVs isolated from GJ of GC patients and non-cancer individuals. (**A**) An example of an electropherogram of small RNA from Agilent 2100 Bioanalyzer. (**B**) The relative expression of miR-451a, miR-199a-3p, miR-135b-3p (* *p* < 0.05); miR-204-3p and miR-135b-5p (*p* > 0.05) in EVs of GC patients (Tumor) and non-cancer individuals (Normal) from RT-qPCR data. Gene expression data were normalized to miR-23a. Fold change (FC) was determined using the ΔΔCt method.

## Data Availability

The data presented in this study are available in the article and Appendix A.

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
