# Peer review of "Isolation and Characterization of Extracellular Vesicles from Gastric Juice"

_cancers, 2022, doi:10.3390/cancers14143314_

Round 1

Reviewer 1 Report

Skryabin et al. studied the "Isolation and characterization of extracellular vesicles from gastric juice". Even though EVs are considered important biomarkers for different diseases, I couldn't find sufficient novelty in this study. The authors claimed that "To our surprise, EVs from GJ have been largely unexplored." which is not true at all. There are many studies already reported recently. Extensive English editing is also required. The specific comments are below.

Abstract needs revision. Mention the objective clearly and write a summary of your results and specific conclusion. 

The introduction needs extensive revision as it failed to discuss the problem statement clearly and why this study is important and needs to evaluate EVs as the same study has been conducted by Kagota et al. 2019 using the same samples, GJ from GC (https://doi.org/10.3390%2Fijms20040953). Also, the authors did not mention what objective they are going to achieve. Some other statements are not true (mentioned below). Correct those carefully. 

Line 48-60: Remove these sentences. 

Line 70: "Gastric cancer is one of the most common cancers and the second leading cause of cancer-related death worldwide [11]". This data is too old data. GC is not 2nd leading cancer-related death anymore. Currently, CRC is the second leading cause of death. You can get the latest CRC epidemiology at https://doi.org/10.3390/cancers14071732. 

Line 102-103: "However, the authors of the latter study failed to obtain EV preparations using standard ultracentrifugation methodology, and developed their own rather complicated preprocessing procedure for EV isolation." Why did they fail to isolate EVs? How does your study overcome its shortcomings? Did you discuss it sufficiently in your study? Unfortunately, I couldn't find such information. 

Line 108: "(intermediate to high grade adenocarcinoma...)" Give the grading data as a supplementary file. Have you checked the case-to-case variation of EVs? It's important for your study. I couldn't find this information in statistical analysis and results too. 

What method did you follow for sample processing? You did not mention it. 

You mentioned that you isolated EVs following [25] but I found there are changes in the protocol. What is your justification for the changes? Why did you use the first centrifugation for 3 hr? You also used a higher speed. Have you checked the differences between EVs for different centrifugation speeds? 

Line 101: "In 2019 Kagota et al. clarified the existence of EVs in GJ for the first time [24]". Did Kagota et al claim their study as for the first time? I don't think so. Their study also mentioned, "......analysis of GJ-EVs has not been adequate, with only a few reports about them [25]". It means GJ-EVs are already studied but not adequate yet. So, be sure to state such a claim.

Line 322-324 No need as mentioned in the method. But write the findings directly that you got from this technique. 

Figures 2 and 3. Mentioned what refers to T1, N1,..... etc. Don't mention only sample T1. Write it clearly what actually T1 sample is. 

Line 367-368: Cite "numerous" articles appropriately that were published recently. 

Line 387: "clinical specimens" cite this as well.

Line 559-560: Add specific criteria of GJ-EVs in conclusion that can be used as GC markers. 

Reviewer 2 Report

Authors have high similarity in the article with textual sections of an article published by Skryabin et al. in March 2022 https://doi.org/10.3390/cells11071064 (reference 66). Therefore, it is crucial to rewrite and cite the article in the methods sections.

Round 2

Reviewer 1 Report

Even though I do not agree with all points described in the authors' response, but authors made significant improvements in the revision. It can be accepted after moderate English editing. 

Reviewer 2 Report

I don't have any comments; the authors made sufficient changes to the manuscript 

This manuscript is a resubmission of an earlier submission. The following is a list of the peer review reports and author responses from that submission.